# Effects of Fatty Acid Metabolites on Adipocytes Britening: Role of Thromboxane A2

**DOI:** 10.3390/cells12030446

**Published:** 2023-01-30

**Authors:** Cécilia Colson, Pierre-Louis Batrow, Sebastian Dieckmann, Laura Contu, Christian H. Roux, Laurence Balas, Claire Vigor, Baptiste Fourmaux, Nadine Gautier, Nathalie Rochet, Nathalie Bernoud-Hubac, Thierry Durand, Dominique Langin, Martin Klingenspor, Ez-Zoubir Amri

**Affiliations:** 1Université Côte d’Azur, CNRS, Inserm, iBV, 06107 Nice, France; 2Chair for Molecular Nutritional Medicine, Technical University of Munich, TUM School of Life Sciences, 85354 Freising, Germany; 3Rheumatology Department, Hospital Pasteur 2 Centre Hospitalier Universitaire (CHU), 06000 Nice, France; 4Institut des Biomolécules Max Mousseron, Pôle Chimie Balard Recherche, UMR 5247, CNRS, University Montpellier, 34093 Montpellier, France; 5Université de Lyon, INSA Lyon, CNRS, Laboratoire de Mécanique des Contacts et des Structures (LaMCoS), UMR5259, 69621 Villeurbanne, France; 6Institute of Metabolic and Cardiovascular Diseases, I2MC, University of Toulouse, INSERM, University of Toulouse III—Paul Sabatier (UPS), 31400 Toulouse, France; 7Laboratoire de Biochimie, Centre Hospitalier Universitaire de Toulouse, 31000 Toulouse, France; 8Institut Universitaire de France (IUF), 75231 Paris, France

**Keywords:** brown adipocyte, white adipocyte, conversion, PGI2, TXA2, FAHFA, 15dPGJ2, 15dPGJ3

## Abstract

Obesity is a complex disease highly related to diet and lifestyle and is associated with low amount of thermogenic adipocytes. Therapeutics that regulate brown adipocyte recruitment and activity represent interesting strategies to fight overweight and associated comorbidities. Recent studies suggest a role for several fatty acids and their metabolites, called lipokines, in the control of thermogenesis. The purpose of this work was to analyze the role of several lipokines in the control of brown/brite adipocyte formation. We used a validated human adipocyte model, human multipotent adipose-derived stem cell model (hMADS). In the absence of rosiglitazone, hMADS cells differentiate into white adipocytes, but convert into brite adipocytes upon rosiglitazone or prostacyclin 2 (PGI2) treatment. Gene expression was quantified using RT-qPCR and protein levels were assessed by Western blotting. We show here that lipokines such as 12,13-diHOME, 12-HEPE, 15dPGJ2 and 15dPGJ3 were not able to induce browning of white hMADS adipocytes. However, both fatty acid esters of hydroxy fatty acids (FAHFAs), 9-PAHPA and 9-PAHSA potentiated brown key marker UCP1 mRNA levels. Interestingly, CTA2, the stable analog of thromboxane A2 (TXA2), but not its inactive metabolite TXB2, inhibited the rosiglitazone and PGI2-induced browning of hMADS adipocytes. These results pinpoint TXA2 as a lipokine inhibiting brown adipocyte formation that is antagonized by PGI2. Our data open new horizons in the development of potential therapies based on the control of thromboxane A2/prostacyclin balance to combat obesity and associated metabolic disorders.

## 1. Introduction

The increasing prevalence of overweight and obesity has reached “epidemic” proportions with over two billion people overweight (BMI > 25 kg/m^2^) and at least 650 million of them clinically obese (BMI > 30 kg/m^2^) [1]. Obesity represents a risk factor for hypertension, type 2 diabetes and cardiovascular diseases. This increase in body weight, accompanied by an increase in the mass of adipose tissue, results from an imbalance between energy intake and energy expenditure. So far, pharmacological remedies or lifestyle interventions normalizing this imbalance with significant long-term success are not efficient; for morbidly obese patients, no effective treatment other than bariatric surgery is available.

The adipose organ, composed of white (WAT) and brown adipose tissue (BAT), plays a central role in the control of energy homeostasis [2,3,4]. In contrast to WAT, BAT is specialized in adaptive thermogenesis in which the uncoupling protein 1 (UCP1) plays a key role [5,6]. It is well established that active brown and brite (or beige) adipocytes are present in healthy adults and hold great promise as a novel antiobesity strategy in humans [7,8,9,10,11,12]. Cold exposure is the most potent physiologic stimulus of BAT thermogenesis that results in increased energy expenditure in mice and humans. Several compounds have been reported to promote brite adipogenesis or white to brite conversion [13,14,15,16].

A number of lipids are known to play key roles in cell signaling, homeostasis and disease. They are substrates and products of enzymes constituting metabolic pathways. Lipids derive directly from diet or de novo synthesis from precursors and their metabolism is under the control of environmental and genetic variation. Besides serving as fuel source for heat generation, certain lipids called lipokines might play a crucial role in controlling the differentiation and activation of thermogenic adipocytes [14,15,17]. Interestingly, 12,13-diHOME and 12-HEPE have recently been described to be strongly correlated with BAT activity [18,19,20]. Prostaglandins and prostacyclins, as well as the recently discovered family of fatty acid esters of hydroxy fatty acids (FAHFAs), also represent potential effectors of white into brown adipose tissue conversion and are of particular interest since their bioavailability can be modulated by nutrition [21,22]. Furthermore, arachidonic acid metabolites, such as thromboxane, modulate adipocyte differentiation [23,24]; however, their involvement in the control brown adipocytes formation and function remains to be elucidated.

In the present study, we analyzed the involvement of fatty acid metabolites towards brite adipocyte formation and function. To this end, we took advantage of a unique human cell model, namely the human multipotent adipose-derived stem [25,26] cells, which can undergo adipocyte differentiation process and convert into functional brite thermogenic adipocytes upon stimulation with rosiglitazone, a potent PPARγ agonist [26,27,28]. This trans-differentiation of white into brown adipocytes is characterized by a metabolic reprogramming and morphological modifications [26,27,28]. We tested various compounds either selected from the literature (FAHFAs, prostaglandins and metabolites) or identified in our previous correlative study ([29] and unpublished data). Most of the compounds tested were not efficient in the control of brite adipocyte formation, except two FAHFAs, which modestly potentiated rosiglitazone-induced browning. However, we discovered an antagonism of thromboxane A2 and PGI2 in the control of white adipocyte browning.

## 2. Materials and Methods

### 2.1. Reagents

Cell culture media, insulin and trypsin buffers were purchased from Invitrogen, fetal bovine serum was purchased from Eurobio (Les Ulis, France), hFGF2 was purchased from Peprotech (Neuilly Sur Seine, France) and other reagents were purchased from Sigma-Aldrich Chimie (Saint-Quentin Fallavier, France). All tested bioactive fatty acid compounds were purchased from Cayman. 15-deoxy-Delta(12,14)-prostaglandin J3 was synthetized from PGD3 as previously described [30]. FAHFAs were synthetized, as previously described [31], or purchased from Cayman (Montigny-le-Bretonneux, France). All compounds were dissolved in DMSO, purged with argon as recommended by manufacturers and checked by MS/MS before their use in culture.

### 2.2. Cell Culture

The establishment and characterization of hMADS cells (human multipotent adipose-derived stem) have previously been described [25,32]. Cells were seeded at a density of 5000 cells/cm2 in Dulbecco’s Modified Eagle’s Medium (DMEM) supplemented with 10% FBS, 15 mM Hepes, 2.5 ng/mL hFGF2, 60 mg/mL penicillin and 50 mg/mL streptomycin. hFGF2 was removed when cells reached confluence. Cells were induced to differentiate in a serum-free medium at day 2 post-confluence (designated as day 0) in DMEM/Ham’s F12 (1:1) media, supplemented with 10 µg/mL transferrin, 10 nM insulin, 0.2 nM triiodothyronine, 1 µM dexamethasone and 500 µM isobutyl-methylxanthine for 4 days. Cells were treated between day 2 and 9 with 100 nM rosiglitazone (a PPARγ agonist), to enable white adipocyte differentiation. At day 14, conversion of white to brite adipocytes was induced by rosiglitazone or the indicated compound for 4 days. Media were changed every other day and cells were used at the indicated days. Fatty acid metabolites were added to culture media in the presence of 5 µM bovine serum albumin.

### 2.3. Adiponectin Levels

hMADS cells were differentiated into white adipocytes and induced to convert into brite adipocytes in the absence or presence of 15Delta-PGJ2 and 15Delta-PGJ3. Media were collected 6 h after the last medium change, centrifuged and frozen at −80 °C. Adiponectin levels were measured in media using an ELISA kit as per the manufacturer’s instructions (Invitrogen, Cergy Pontoise, France.

### 2.4. Isolation and Analysis of RNA

These procedures followed MIQE standard recommendations and were conducted as previously described [33]. The oligonucleotide sequences, designed using Primer Express software, are shown in Appendix A. Quantitative PCRs (qPCRs) were performed using SYBR qPCR premix Ex TaqII from Takara (Ozyme, Saint Cyr L’Ecole, France) and assays were run on a StepOne Plus ABI real-time PCR machine (Perkin Elmer Life and Analytical Sciences, Boston, MA, USA). The expression of selected genes was normalized to that of 36B4 housekeeping gene and then quantified using the comparative-ΔCt method.

### 2.5. Western Blot Analysis

Proteins were extracted from cells or tissues as previously described [34]. Equal amounts of cellular proteins, 30 to 50 µg, were separated by electrophoresis using gradient gels (4–15%) and blotted onto PVDF membranes. Following blocking, membranes were incubated. Primary antibody incubation was performed overnight at 4 °C (anti-UCP1, Abcam #ab10983, dilution 1:1000; anti-TBP, CST #D5C9H, dilution 1:1000). Primary antibodies were detected with HRP-conjugated anti-rabbit or anti-mouse immunoglobulins (Promega, Charbonniere Les Bains, France). Detection was performed using Immobilon Western Chemiluminescent HRP Substrate (Merk-Millipore, Fontenay Sous Bois, France). Chemiluminescence obtained after adding Pierce ECL Western blotting substrate (Thermo Scientific, Asnièrse sur Seine, France) was detected using an Amersham Imager 600 and quantified with Image Lab 5.0 software (Bio-Rad, Marnes-la-Coquette, France).

### 2.6. Statistical Analyses

Data are expressed as mean values ± SEM and were analyzed using InStat software (GraphPad Prism version 8.3.0 for Windows, GraphPad Software, San Diego, California, USA). Data were analyzed by Student’s t-test or one-way ANOVA followed by a Dunett’s multiple comparisons test or two-way ANOVA followed by Tukey’s multiple comparisons test. Differences were considered statistically significant when *p* < 0.05.

## 3. Results

### 3.1. 9-PAHSA and 9-PAHPA Effects on Browning of White hMADS Adipocytes

Lipidomic analysis of adipose tissues revealed the existence of a novel class of branched fatty acid esters of hydroxy fatty acids (FAHFAs) that are endowed with anti-inflammatory and anti-diabetic properties [21,35,36,37]. We aimed to test the effects of the most characterized FAHFAs in the process of white adipocyte browning. We investigated the effect of 10 different FAHFA compounds using hMADS cell model, using various amounts of FAHFA (between 0 to 10 µM).

Cells were induced to differentiate into white adipocytes for 14 days and then induced to convert into brite adipocytes for 4 days in the presence of rosiglitazone. The effect of different FAHFAs on brite adipocyte formation in the absence (white) or presence (brite) of rosiglitazone was assessed by gene expression analysis of thermogenic (UCP1, Figure 1, left panel) and adipogenic (FABP4, Figure 1, right panel) markers.

Palmitic acid 9-hydroxypalmitic acid (9-PAHPA) and palmitic acid 9-hydroxystearic acid (9-PAHSA) treatment showed no increase in UCP1 mRNA levels in white adipocytes, but 1 µM of 9-PAHPA induced a slight increase in UCP1 expression in the presence of rosiglitazone. A slight increase was also obtained in FABP4 mRNA expression with 1 µM of 9-PAHSA; 9-PAHPA induced a decrease in white adipocytes.

Palmitic acid 5-hydroxystearic acid (5(R)-PAHSA) treatment also showed no effect on UCP1 mRNA levels in white or brite adipocytes (Figure 1, left panel). However, it induced a slight significant decrease in the levels of adipogenic marker FABP4 (Figure 1, right panel).

Other FAHFAs, such as 7-PAHSA, 5-PAHPA, 5-OAHPA and 9-OAHPA, showed no effect on UCP1 and FABP4 mRNA levels in white and brite adipocytes (data not shown).

### 3.2. Effects of Prostaglandins and Their Derivatives on Browning of White Adipocytes

We and others have shown that prostacyclin 2 (PGI2), through the use of a stable compound carbaprostacyclin (cPGI2), induced brown adipocyte formation and function [38,39,40]. Herein, we aimed to test whether the polyunsaturated fatty acids, including omega-6 and omega-3-derived prostanoids (PGI2 and PGI3) and their metabolites ((15-deoxy-delta 12,14-PGJ2 (15dPGJ2) and 15-deoxy-delta 12,14-PGJ3 (15dPGJ3)) affected the browning process of white adipocytes.

White hMADS adipocytes obtained after 14 days of culture as described above, were treated with PGI3, 15dPGJ2 or 15dPGJ3 from day 14 to day 18. Treatments with cPGI2 or 100 nM rosiglitazone were used as positive controls. UCP1 mRNA levels were analyzed at day 18 as a marker of brite adipocyte conversion. In agreement with previous reports [38], UCP1 gene expression increased in cPGI2-treated cells compared to untreated cells in a dose-dependent manner, though less efficiently than 100 nM rosiglitazone (Figure 2, left panel). PGI3 treatment had no effect on UCP1 gene expression. cPGI2 and rosiglitazone also enhanced the expression of FABP4 as expected, whereas no effect was observed with PGI3 (Figure 2, middle panel). Finally, PLIN1 a standard marker of adipogenesis was not affected by any of these compounds (Figure 2, right panel).

Then, we tested whether 15dPGJ2 and 15dPGJ3, which were previously reported as potential PPARγ ligands [30,41], were able to induce brite adipocyte formation. In contrast to the strong effect of rosiglitazone, neither compound induced UCP1 gene expression (Figure 3A). 15dPGJ2 and 15dPGJ3 showed a tendency toward a slight increase in FABP4 gene expression compared to the strong effect of rosiglitazone (Figure 3B). As expected, adiponectin gene expression was increased by rosiglitazone; however, 15dPGJ3 has no effect, and a slight but not significant effect of 15dPGJ2 (Figure 3D) was observed. Of note, PLIN1 gene expression was not affected under any of these conditions (Figure 3C). In parallel, we analyzed whether adiponectin secretion was affected. Brite adipocytes secreted a higher amount of adiponectin compared with white adipocytes, but neither 15dPGJ2 nor 15dPGJ3 altered adiponectin secretion (Appendix A).

### 3.3. Effects of 12,13 di-HOME, 12-HEPE and 18-HEPE on Browning of White Adipocytes

In a previous study, we quantified a panel of 36 fatty acid metabolites in brown and white adipose tissues of mice, in human brown and white adipose tissue biopsies and in hMADS adipocytes in order to correlate their levels to UCP1 mRNA expression ([42] and unpublished data). We aim to analyze the effects of some of these metabolites as well as two compounds (12,13 di-HOME and 12-HEPE) recently identified to play a role in thermogenesis [18,19,20].

We asked whether these fatty acid metabolites can affect the differentiation of brite adipocytes. For this purpose, as described above, hMADS cells were first induced to differentiate into white adipocytes for 14 days, before testing for the next 4 days the potential of these metabolites either to induce the conversion into brite adipocytes or to modulate the rosiglitazone-induced browning.

12,13 di-HOME (Figure 4A) and 12-HEPE (Figure 4B) did not induce UCP1 gene expression (left panels) and did not interfere with rosiglitazone-induced effect (right panel). Both compounds did not affect FABP4 and PLIN1 mRNA levels (Figure 4A,B).

Among the identified compounds were 18-HEPE, 9/13-HODE and 5/12/15-HETE. We observed no effect with these compounds on the expression of white and brite adipocytes markers, as shown for example in 18-HEPE (Figure 4C and data not shown).

### 3.4. Effects of Thromboxane A2 on Browning of White Adipocytes

In a previous study, we analyzed the levels of a set of eicosanoids (36 compounds) in plasma and adipose tissues [29,34]. Among these fatty acid metabolites, thromboxane B2 (TXB2), which reflects the active compound thromboxane A2 (TXA2), was negatively associated with UCP1 expression in murine WAT and BAT [29,34]. We aimed to further analyze its effects on a cellular model. Since TXA2 is rapidly degraded into TXB2, we used its stable analogue, the carbocyclic thromboxane A2 (CTA2). As shown in Figure 5A, CTA2 induced a significant slight increase in the expression of UCP1 mRNA in white adipocytes (3.2 fold increase), whereas UCP1 protein levels were not affected (Figure 5B). However, it significantly inhibited the rosiglitazone-induced UCP1 gene expression. CPT1M, FABP4 and PLIN1 mRNA levels, other thermogenic and adipogenic markers, were not affected by CTA2 treatment in either white or brite adipocytes. TXB2, which represents the inactive product of TXA2, did not affect UCP1 gene expression in white and brite adipocytes (Appendix A). In gene expression data, UCP1 protein level consistently decreased with CTA2 treatment (Figure 5B) in brite hMADS adipocytes with an optimal inhibitory effect of 1 µM of CTA2, whereas no effect was observed with TXB2 treatment (Appendix A).

We aimed to know whether the TXA2 (CTA2) effects were specific to human cells. For this purpose, we carried out similar experiments in mouse primary adipocytes. Stroma-vascular fraction cells from subcutaneous white or brown (Appendix A) adipose tissues of mice were induced to differentiate into brown adipocytes in the presence of rosiglitazone for 7 days, and treated for the last 4 days with various amounts of CTA2. Rosiglitazone treatment led to induction of UCP1 mRNA expression, which was significantly inhibited by CTA2 treatment in a dose-dependent manner in both stroma-vascular fraction cells derived from subcutaneous white and brown adipose tissues (Appendix A). The mRNA level of other thermogenic markers FABP4 and CPT1M was not affected while PLIN1 mRNA expression increased at the higher dose, 10 µM of CTA2.

Altogether, these results show that CTA2 inhibits the expression of the UCP1 gene in human and mouse adipocytes.

During the inflammation process, TXA2 is reported to antagonize PGI2 effects in terms of vasodilation and platelet aggregation [43]. In order to further characterize the effects of TXA2, we aimed to know whether TXA2 antagonizes PGI2 effect to control the browning process. As expected, cPGI2 (3 µM) increased mRNA expression of UCP1. cPGI2-induced UCP1 mRNA expression decreased with CTA2 treatment in a dose-dependent manner (Figure 6, left panel). mRNA level of CPT1M was also significantly decreased with a higher dose of CTA2 (10 µM), while FABP4 mRNA level was not affected (Figure 6, middle and right panels).

PGI2 and TXA2 exert their effects through membrane receptors PTGIR and TXA2R, respectively [44,45]. Both receptors are expressed in hMADS adipocytes (Figure 7A,B) and significantly reduced in the presence of 100 nM rosiglitazone. The addition of cPGI2 tends to reduce mRNA levels of PTGIR, mimicking the effects of rosiglitazone.

To further study TXA2R’s role in CTA2 inhibitory effect, we used a specific receptor antagonist (L655,240) in the presence of rosiglitazone (Figure 7C) or cPGI2 (Figure 7D). The results showed that inhibition of UCP1 mRNA expression by CTA2 was partially reversed in the presence of L655,240, suggesting the involvement of other pathways in CTA2 effects.

## 4. Discussion

Obesity has become pandemic and the need to restore the energy balance has led to development of new therapeutic strategies. Formation and activation of brown adipose tissue represent a potential target that is being widely studied. Lipokines have been shown to regulate thermogenesis through increased energy expenditure and improved systemic metabolism and may constitute potential therapeutic targets [17,46]. Metabolites of ω6 and ω3 polyunsaturated fatty acids represent signaling molecules involved in the control of adipogenesis, energy balance and for some in the activation and recruitment of brown adipocytes in mice and humans [22]. In a previous study, we analyzed the association between UCP1 mRNA expression and a panel of fatty acid metabolites upon hMADS cell treatment and nutritional manipulation using mice, where few compounds correlated negatively or positively [29,34,42,47].

In this study, using a human cell model, we analyzed the effects of various fatty acid metabolites on the formation of brown/brite adipocytes reported in our previous work and from the literature [20,21,29,34,42,48]. Fatty acid esters of hydroxy fatty acids (FAHFAs) have been discovered as a novel class of endogenous mammalian lipids having anti-inflammatory and antidiabetic effects [49,50,51]. None of the FAHFAs used in this study were able to induce any conversion of white to brite adipocytes, contrasting with previous data showing that 9-PAHSA promotes browning of white 3T3-L1 adipocytes and in vivo using wild type and ob/ob mice [36]. Since FAHFAs act through GPR120, this discrepancy might be explained by a difference between human and mouse models, although hMADS cells express this receptor (data not shown). Nevertheless, 9-PAHPA significantly potentiated UCP1 mRNA levels in rosiglitazone-induced brite adipocytes. This observation is consistent with recent in vivo studies showing that 9-PAHPA long-term intake by mice improves basal metabolism and insulin-sensitivity in healthy and diet-induced obesity mice [36,37,52]. Prostaglandins play an important role in the control of adipogenesis and thermogenesis [17,45]. Our data show that PGI3, derived from the EPA ω3 polyunsaturated fatty acid, is inefficient to induce UCP1 gene expression, in contrast to prostacyclin (PGI2) derived from the ARA ω6 polyunsaturated fatty acid. Previous studies displaying similar anti-aggregatory effects of PGI3 and PGI2 on human and rabbit platelets have been reported, suggesting that these compounds can exert either similar or different effects [53]. These discrepancies might be explained by the use of stabilized analog of PGI2 and cPGI2, and not for PGI3. They could also be due to a different model (adipocytes versus platelet), which could express higher levels of IP receptors and/or more efficient signal transduction. Other EPA ω3 prostanoid intermediates, PGD3 and PGJ3 (data not shown), were also not efficient in inducing the expression of thermogenic markers in hMADS white adipocytes.

We further analyzed the effects of metabolites previously reported to act as endogenous natural ligands of PPARγ, the master transcriptional regulator of adipogenesis [41,54]. The PGJ2 and PGJ3 metabolites, 15-deoxy-delta 12,14-PGJ2 and 15-deoxy-delta 12,14-PGJ3, respectively, were inefficient to induce any browning of white hMADS adipocytes. 15-deoxy-delta 12,14-PGJ3 did not induce adiponectin secretion in contrast to what was reported in a mouse cell model [30]. Further studies are necessary to clarify the role of 15-deoxy-delta 12,14-PGJ3 in adipogenesis as few experiments are reported. In the absence of effects of these prostaglandins and their derivatives on UCP1 expression, it remains important to analyze other physiological processes such as lipolysis, oxygen consumption or glucose transport.

Among other oxylipins, 12,13-diHOME, 12-HEPE and 18-HEPE failed to induce browning process of white adipocytes. However, thromboxane A2 (TXA2) inhibited the rosiglitazone-induced expression of UCP1 mRNA and protein. We found that levels of thromboxane B2 (TXB2), an inactive metabolite of TXA2, are negatively associated with UCP1 expression in our previous study ([29] and unpublished data). TXA2 is a chemically unstable lipid mediator involved in several pathophysiologic processes. In human platelets, TXA2 is the major arachidonic acid derivative found via the cyclooxygenase (COX1) pathway. Assessment of TXA2 biosynthesis can be performed through measurement of serum TXB2 [55]. The use of CTA2, a stable analogue of TXA2, allowed us to show that TXA2 plays an anti-browning effect, probably favoring whitening of brown adipocytes. TXA2 exerted its effects through thromboxane A2 receptor (TXA2R), which is expressed in hMADS adipocytes. Of note, PTGIR (PGI2 receptor) and TXA2R (TXA2 receptor) are expressed by white and brite adipocytes and the inhibitory effects of CTA2 were partially reversed in the presence of a selective TXA2R antagonist. These results imply that inhibitory effects of TXA2 are controled by several pathways, which might be related to the previously described PGI2 indirect effect on thermogenic phenotype acquisition through its receptor and direct PPARγ activation [38]. Thromboxane A2 is an arachidonate metabolite that is a potent stimulator of platelet aggregation, a constrictor of vascular and respiratory smooth muscles and is considered a mediator in diseases such as myocardial infarction, stroke and bronchial asthma [43]. It has been reported that TXA2 modulates peripheral tissue insulin sensitivity and adipose tissue fibrosis [56], which could explain the loss of thermogenic phenotype in our human cell model. Furthermore, it has been shown that TXB2 (reflecting TXA2) circulating levels were higher in obese compared to lean patients [57]. COX catalyzes the first step in the conversion of arachidonic acid into prostanoids, which includes prostacyclin (PGI2) and TXA2 that have opposite effects on cAMP signaling in platelets, vasomotor regulation, blood pressure and platelet aggregation. PGI2 increases cAMP, vasodilates blood vessels, lowers blood pressure and inhibits platelet aggregation, whereas TXA2 opposes these actions [58]. Given this, PGI2 favors adipogenesis and thermogenesis, whereas TXA2 opposes these effects and resembles their antagonistic effect on platelet aggregation. PGI2 and TXA2 derive from the same precursor, arachidonic acid, through the action of the two cyclooxygenases isoforms (COX1 and 2), giving rise to a common substrate (PGH2). PGH2 is then metabolized by enzymes, specifically prostaglandin I synthase or thromboxane synthase, to produce PGI2 or TXA2, respectively. It is tempting to speculate that inhibiting the production of TXA2 locally or developing a specific TXA2 receptor antagonist in association with increasing PGI2 levels will favor brite and brown adipocyte formation through trans-differentiation of white adipocytes into brown adipocytes and function. Moreover, recent findings show that coagulation factors [59] can mediate biological effects in BAT that may suggest a link to the involvement of thromboxane. More detailed studies, both in vitro and in vivo, are necessary to decipher the mechanisms behind these effects. Our data open new horizons in the development of potential therapies based on the control of thromboxane A2/prostacyclin balance to combat obesity and associated metabolic disorders.

## 5. Conclusions

In conclusion, in search of compounds controlling brown adipocyte formation, we found that thromboxane A2 exhibited an inhibitory effect on the conversion of white adipocyte into brown adipocyte, i.e., browning process, which is antagonized by PGI2. Our data revealed an important role of TXA2 and PGI2 in the remodeling of the adipose organ. We also found that 9-PAHSA and 9-PAHPA modestly potentiate rosiglitazone-induced browning of white adipocytes.

## Figures and Tables

**Figure 1 cells-12-00446-f001:**
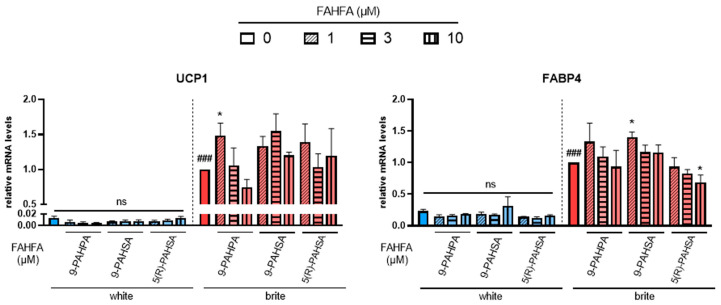
Effects of FAHFA on UCP1 mRNA expression. White hMADS adipocytes (blue columns) were converted (from day 14 to day 18) into brite adipocytes (red columns) using the PPARγ agonist rosiglitazone in the absence (empty) or the presence of 1, 3 or 10 µM (shades) 9-PAHSA, 9-PAHPA and 5(R)-PAHSA. mRNA levels of UCP1 and FABP4 markers were measured. Histograms display mean ± SEM of three to five independent experiments; statistics were conducted through a one-way ANOVA with Dunett’s multiple comparisons test, where *p* < 0.05 was considered significant: ###: *p* < 0.001, white vs. brite adipocyte; *: *p* < 0.05, treated vs. untreated white or brite adipocytes.

**Figure 2 cells-12-00446-f002:**
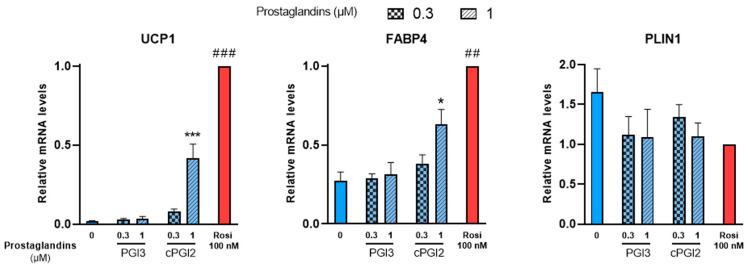
PGI3 does not induce thermogenic markers, unlike cPGI2. White hMADS adipocytes (blue columns) were induced to convert (from day 14 to day 18) into brite adipocytes (red comumns) in the presence of various amounts of PGI3, using cPGI2 and the PPARγ agonist rosiglitazone (Rosi 100 nM) as positive controls. mRNA levels of UCP1, FABP4 and PLIN1 markers were measured. Histograms display mean ± SEM of three to six independent experiments; statistics were conducted through a one-way ANOVA with Dunett’s multiple comparisons test, where *p* < 0.05 was considered significant: *: *p* < 0.05, ***: *p* < 0.001, treated vs. untreated; ##: *p* < 0.01, ###: *p* < 0.001, vs. 1 µM cPGI2 treatment.

**Figure 3 cells-12-00446-f003:**
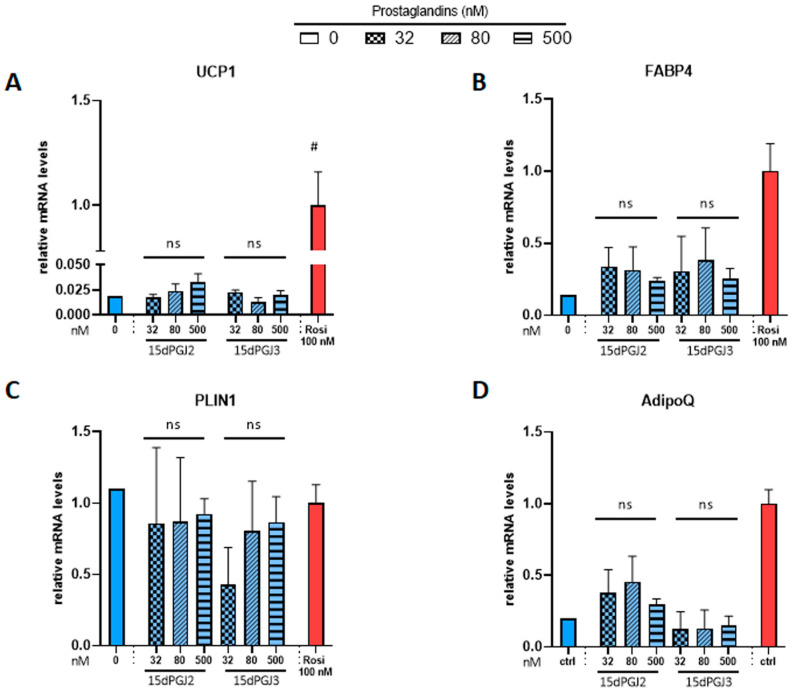
Potential PPARγ activators, 15-deoxy-delta 12,14-PGJ2 and 15-deoxy-delta 12,14-PGJ3, don’t influence mRNA levels of PPARγ target genes. White hMADS adipocytes (blue comumns) were induced to convert (from day 14 to day 18) into brite adipocytes (red columns) in the presence of various amounts of 15-deoxy-delta 12,14-PGJ2 (15dPGJ2) or 15-deoxy-delta 12,14-PGJ3 (15dPGJ3) using rosiglitazone (Rosi 100 nM) as positive control. mRNA levels of UCP1 (**A**), FABP4 (**B**), PLIN1 (**C**) and adiponectin (**D**) markers were measured. Histograms display mean ± SEM of two to eight independent experiments; Statistics were conducted through a one-way ANOVA with Dunett’s multiple comparisons test, where *p* < 0.05 was considered significant: #: *p* < 0.05, white control (0) vs. brite control (Rosi 100 nM) adipocyte. ns: not significant.

**Figure 4 cells-12-00446-f004:**
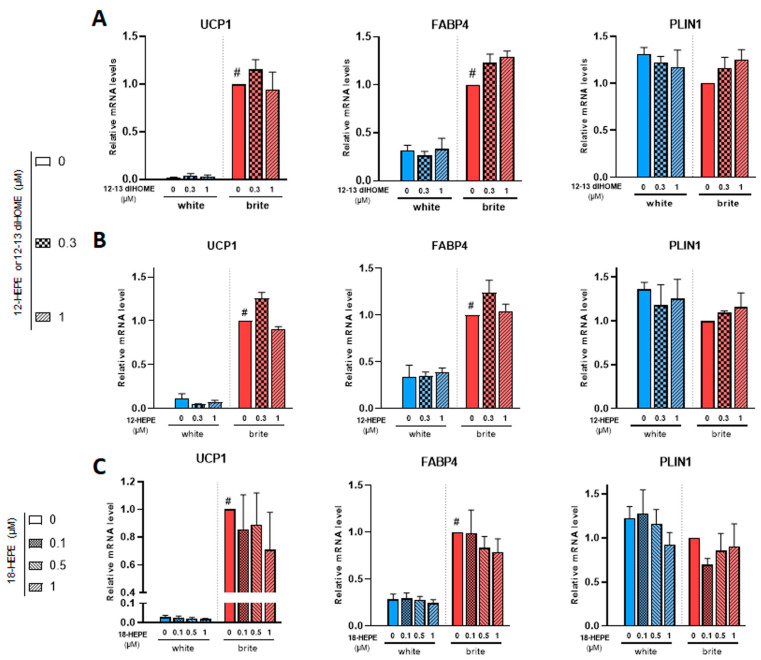
Effects of 12,13 diHOME, 12-HEPE and 18-HEPE on UCP1 mRNA expression. White hMADS adipocytes (blue columns) were induced to convert (from day 14 to day 18) into brite adipocytes (red columns) in the presence of various amounts of 12,13 diHOME (**A**), 12-HEPE (**B**) or 18-HEPE (**C**) in the absence (white) or the presence of 100 nM rosiglitazone (brite). UCP1, FABP4 and PLIN1 mRNA levels were measured. Histograms display mean ± SEM of three to nine independent experiments; statistics were conducted through a two-way ANOVA with Tukey’s multiple comparisons test, where *p* < 0.05 was considered significant: #: *p* < 0.05, white (blue) vs. brite (red) adipocyte.

**Figure 5 cells-12-00446-f005:**
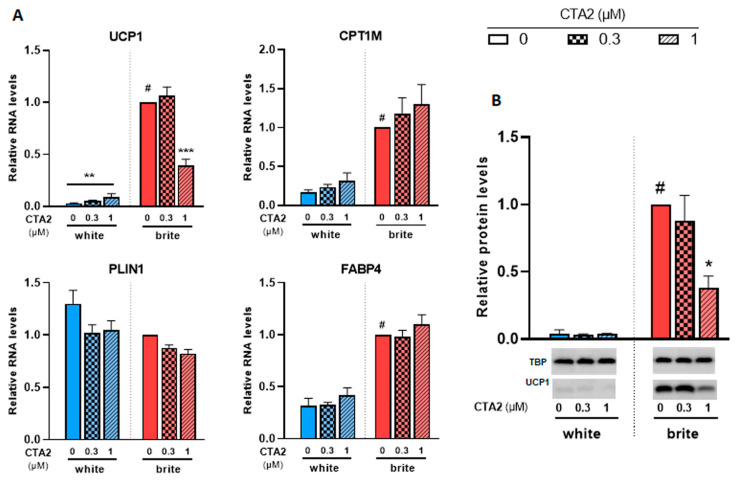
CTA2 inhibits UCP1 expression. White hMADS adipocytes (blue columns) were induced to convert (from day 14 to day 18) into brite adipocytes (red columns) in the presence of various amounts of CTA2 in the absence (white) or the presence of 100 nM rosiglitazone (brite). UCP1, CPT1M, PLIN1 and FABP4 mRNA levels were measured (**A**). 40 µg total protein extracts were analyzed by Western blot (**B**). Histograms display mean ± SEM of three to four (**A**) or two (**B**) independent experiments; statistics were conducted through a two-way ANOVA with Tukey’s multiple comparisons test, where *p* < 0.05 was considered significant: #: *p* < 0.05, white vs. brite adipocyte; *: *p* < 0.05, **: *p* < 0.01, ***: *p* < 0.001, treated vs. untreated.

**Figure 6 cells-12-00446-f006:**
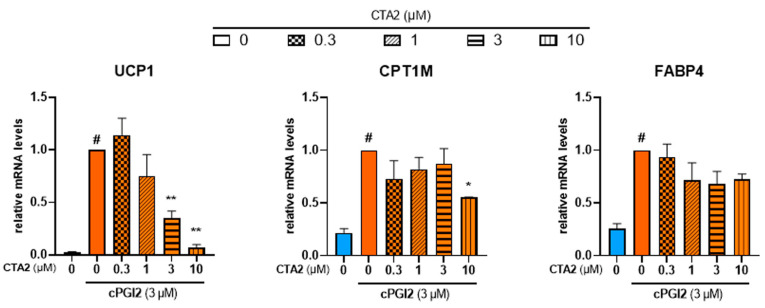
CTA2 limits cPGI2-induced browning process of white hMADS adipocytes. White hMADS adipocytes (blue columns) were induced to convert (from day 14 to day 18) into brite adipocytes (orange columns) in the presence of 3 µM cPGI2 and in absence or presence of various amounts of CTA2. mRNA levels of UCP1, FABP4 and CPT1M were measured. Histograms represent % of mRNA level relative to control condition (3 µM cPGI2) and display mean ± SEM of three to six independent experiments. Statistics were conducted through a one-way ANOVA with Dunett’s multiple comparisons test, where *p* < 0.05 was considered significant: #: *p* < 0.05, white vs. brite adipocyte, *: *p* < 0.05, **: *p* < 0.01, vs. 3 µM cPGI2 treated adipocytes.

**Figure 7 cells-12-00446-f007:**
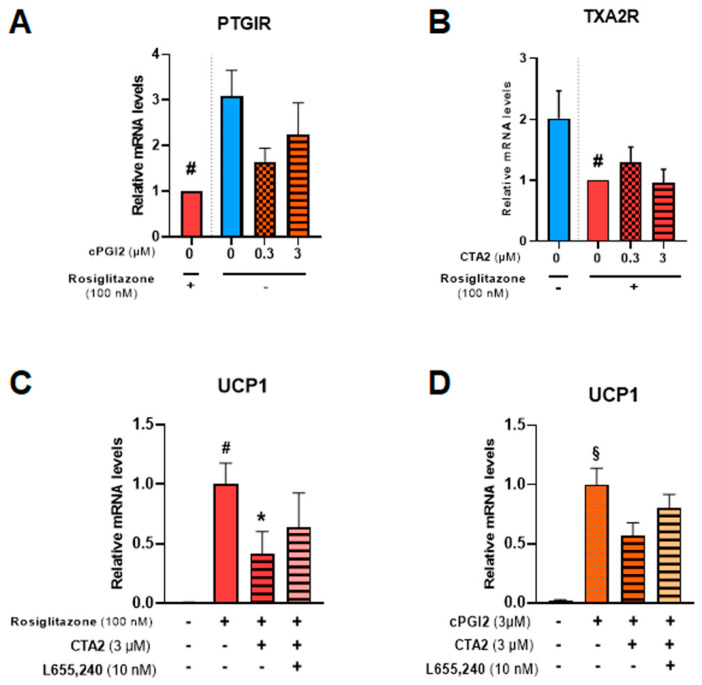
CTA2 effects are partially mediated through its receptor. White hMADS adipocytes (blue columns) were induced to convert (from day 14 to day 18) into brite adipocytes in the presence of cPGI2 (orange columns) or rosiglitazone (red columns), and supplemented or not with 3 µM CTA2, or 10 nM TXA2 receptor antagonist L655,240, as indicated. mRNA levels of PTGIR (cPGI2 receptor) (**A**), TXA2R (CTA2 receptor) (**B**) and UCP1 (**C**, **D**) were analyzed. Histograms display mean ± SEM of three to eight independent experiments. Statistics were conducted through a one-way ANOVA with Dunett’s multiple comparisons test, where p<0.05 was considered significant: *: *p* < 0.05, treated vs. untreated, #: *p* < 0.05, Control vs. rosiglitazone (100 nM), §: *p* < 0.05, Control vs. cPGI2 (3 µM).

## Data Availability

All the data relevant to the present study are available on request from the corresponding author.

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
