# Peer review of "Effects of Fatty Acid Metabolites on Adipocytes Britening: Role of Thromboxane A2"

_cells, 2023, doi:10.3390/cells12030446_

Round 1

Reviewer 1 Report

The study carried by Colson et al. and performed in Dr. Zoubir's lab is a cohmprensive analysis of the effect of several compounds (i.e lipokines) on adipocytes browning. The Authors have designed the experiment in the search of specific compounds with "browning" properties or with potential inhibitory effects. They found that thromboxane A2 exhibit an inhibitory effect on the conversion of white into brown adipocyte and it is antagonized by PGI2. Overall, the data presented reveled an important role of TXA2 and PGI2 in modulating the phenotype of adipocytes. The Authors also identified that 9-PAHSA and 9-PAHPA have the capability of potentiate rosiglitazone-induceed browning of white adipocytes.

The study is well designed and the results obtained are described clearly without speculations and overating it in the discussion section.

To this Reviewer the paper is ready to be accepted after minor modifications maily related to the graphical presentation of the results. In fact, it is not easy to catch the different conditions of treatments in the two main experimental groups (white and brite adipocytes). The Authors should find a way to improve the presentation of the results perhaps assigning colors to the istograms and using shades of the same color to the different concentrations tested. 

Another suggestion is to show, if possible, the morphological appeareance of the cells in the experiment of Figure 5. Based on the relative protein expression (UCP1) it could be of effect to show an immunofluorescence of representative white (as negative control) and brite cells after the treatment with increasing dose of CTA2. Of course this Reviewer is aware that this specific experiment would not chance the message and it is somehow a confirmation of the WB but on the other hand would help the Readers to visualize the morphology of the the cellular model used for all the experiments. 

Author Response

We thank Reviewer 1 for positive comments

Reviewer 2 Report

The manuscript submitted by Colson and colleagues entitled Effects of fatty acid

metabolites on adipocytes britening: role of 2 thromboxane A2 describes the involvement

of several lipokines in white and brite adipocytes formation based on the human Multipotent

Adipose-Derived Stem cell model (hMADS) and genes/protein expression analyzes.

Authors observe that many of tested compound had not effect on the adipocytes browning using

the expression of UCP1 mRNA as a hallmark of that process. However, the results for CTA2,

the analog of thromboxane A2, revealed its role in the inhibition of adipocytes browning induced

by rosiglitazone and prostacyclin 2. In my opinion, this manuscript is composed of two parallel

parts, 1 / devoted to screening of many fatty acid metabolites to determine whether any influence

the browning of white adipocytes, and 2 / the examination of CTA2 effects on browning and

an attempt to explain the mechanism. Although the latter seems to be interesting and introduces

a novelty to the field, the first one could be removed, or changed, introducing only significant

molecules, or moved to the Supplementary Materials. Furthermore, the side experiments

provided on mice adipocytes, however valuable, cause confusion and may be considered to

be moved to the Supplementary Materials. Thus, I recommend reconsidering the change

in the profile and construction of the manuscript.

The submitted manuscript needs to be revised. Some shortcomings have to be corrected

and weak points for which I suggest some modifications listed below. I also have some questions

for which explanations should be included in the manuscript.

The introduction does not correspond to the title as there is no information on current

knowledge of TXA2 in adipocytes britening. Authors describe numerous lipokines, based mostly

on their paper and unpublished data, but no details on i.e. effects and mechanism were provided.

Moreover, each first paragraph in Results section starts with description that should be moved

to the Introduction.

In the experimental section, there is no information about mice adipocytes and location

from the brown adipose tissue were dissected. How were the concentrations of lipokines chosen?

There are also some minor comments:

1/ I suggest to introduce the legend of the groups in figures rather than putting them

in the caption of the figure.

2/ In lines 198 and 304 the references are needed.

3/ In figure 3, the units of concentration are missing.

4/ Line 201: ‘15dPGJ2 and 15dPGJ3 showed a tendency to a slight increase of FABP4

gene expression compared to the strong effect of rosiglitazone (Figure 3B).’ In the preceding

experiment, with PGI3 and PGI2, a higher concentration (1µM) was used. What causes

the concentration change?

5/ Why the number of replicates is so spread i.e. mentioned in caption of figure 3?

6/ What is a possible hypothesis for different effects of CTA2 on UCP1 in white

and brite adipocytes?

7/ In line 253 the explanation of the use of the CPT1M gene is needed.

8/ Can Authors explain why a lower dose of cPGI2 better mimics the rosiglitazone effect?

9/ Please, pay more attention to the editorial aspects of the text, i.e. indices (lines 47, 48,

abbreviations of molecules, etc.), italics (lines 66, 405, genes nomenclature, etc.) and capital

letters only for human genes.

I also recommend creating an abbreviations list, which will make it easier for the reader

to follow the description of the results.

In summary, the submitted manuscript has some weaknesses. Due to all of the abovehighlighted aspects, the manuscript in the present form can be recommended for publication

in Cells only after major revisions.

Author Response

We thank Reviewer 2 for constructive comments

Round 2

Reviewer 2 Report

Authors of the resubmitted manuscript answered all of my comments and the responses are reasonable. My greatest concern about the structure of the manuscript was addressed (major comment) however, it was not changed, only the side experiments provided on murine adipocytes, were moved to the Supplementary Materials. In my opinion, this is a form of compromise and I believe in the justification that presenting negative data results makes sense in this case. Authors improved the clarity of the figures and introduced the sample preparation of murine adipocytes. Authors made appropriate corrections in the text, thus, the manuscript can be recommended for publication in the Cells journal.